# An Accelerated Spectroscopic MRI Metabolite Quantification Based on a Deep Learning Method for Radiation Therapy Planning in Brain Tumor Patients

**DOI:** 10.3390/cancers17030423

**Published:** 2025-01-27

**Authors:** Alexander S. Giuffrida, Karthik Ramesh, Sulaiman Sheriff, Andrew A. Maudsley, Brent D. Weinberg, Lee A. D. Cooper, Hyunsuk Shim

**Affiliations:** 1Department of Radiation Oncology, Emory University School of Medicine, Atlanta, GA 30322, USA; alexander.scot.giuffrida@emory.edu (A.S.G.); karthik.ramesh@emory.edu (K.R.); 2Department of Biomedical Engineering, Emory University and Georgia Institute of Technology, Atlanta, GA 30332, USA; 3Department of Radiation Oncology, Miller School of Medicine, University of Miami, Miami, FL 45056, USA; ssheriff@med.miami.edu (S.S.); amaudsley@med.miami.edu (A.A.M.); 4Winship Cancer Institute, Emory University School of Medicine, Atlanta, GA 30322, USA; brent.d.weinberg@emory.edu; 5Department of Radiology and Imaging Sciences, Emory University School of Medicine, Atlanta, GA 30322, USA; 6Department of Pathology, Northwestern University Feinberg School of Medicine, Chicago, IL 60657, USA; lee.cooper@northwestern.edu

**Keywords:** spectroscopic MRI (sMRI), MR spectroscopy, glioblastoma, radiation treatment, deep learning

## Abstract

Glioblastoma is one of the most aggressive forms of brain cancer, for which the accurate targeting of cancerous tissue during radiation therapy is essential for effective treatment. In this study, we developed a deep learning method called NNFit to quickly analyze spectroscopic MRI scans to create treatment plans for glioblastoma patients. This method focuses on measuring the ratios of three key metabolites, choline, creatine, and N-acetyl aspartate (NAA), to identify metabolically active areas of tumor in the brain. The current fitting methods are based on least squares iterative fitting, which is time-consuming and does not scale up to clinical workflows. Our new method, NNFit, significantly reduces this processing time to under one minute while providing comparable accuracy to the widely used fitting method. This faster processing will enable the clinical adoption of spectroscopic MRI and potentially improve outcomes for patients with glioblastoma.

## 1. Introduction

Glioblastomas (GBMs) are highly aggressive primary brain tumors in adults. The standard treatment for GBMs includes the upfront maximal resection of tumors followed by concurrent chemotherapy (temozolomide, TMZ) and radiation therapy (RT), but this approach fails to achieve long-term tumor control, with 70% of patients experiencing recurrence within six months of treatment [1]. Despite aggressive treatment regimens, outcomes remain poor, with median survival times at approximately 16 months [2,3]. Recurrence is largely due to the infiltrative nature of GBM, which is not accurately assessed on clinical MR images, making it impossible to accurately target the entirety of a tumor.

In a typical RT plan, the tumor-infiltrated regions are targeted by treating hyperintense areas on fluid-attenuated inversion recovery (FLAIR) or T2-weighted (T2w) images with an intermediate dose (45–51 Gy) [4]. However, T2/FLAIR hyperintensity is an indeterminate mix of infiltrative tumor and edema. T1-weighted contrast-enhanced MRI (T1w-CE) identifies leaky neovasculature associated with tumor angiogenesis, but it fails to capture tumors that extend beyond the contrast-enhancing border, limiting its utility in high-dose RT planning (60 Gy). A more precise definition of high-risk recurrence areas could facilitate targeted, high-dose RT, potentially improving local control. 

Our team has pioneered an advanced form of 3D spectroscopic imaging that achieves whole-brain coverage and high resolution, which we denote as spectroscopic MRI (sMRI) [5]. sMRI is a robust application of echo-planar spectroscopic imaging (EPSI) that enhances brain tumor management by providing whole-brain maps of abnormal tissue metabolism. Key metabolic signatures of brain tumor infiltration include a reduction in NAA followed by an increase in Cho, suggesting both a disruption in neuronal integrity and high cell membrane turnover. In a pilot trial consisting of 30 patients from three sites (NCT03137888), we used sMRI to target escalated RT to regions where the ratio of Cho/NAA was greater than twice that in healthy tissue (Cho/NAA ≥ 2x), achieving promising outcomes [6]. Since then, sMRI has made its way into additional trials to enhance patient care by guiding RT [7], proton therapy [8], surgical planning [9], the differentiation of pseudo-progression from true progression (NCT06319027), and the monitoring of therapeutic interventions [10].

Metabolite concentration maps are generated by spectral analyses of individual voxels within an EPSI dataset. These maps help define tumor margins, enabling more precise and personalized RT. However, the adoption of sMRI in clinical settings is limited due to the time-intensive data processing required, which can delay critical treatment decisions [11].

Conventional methods for MR spectroscopy quantification utilize iterative least-squares fitting algorithms (e.g., Levenberg–Marquardt optimization) to estimate metabolite concentrations. While these methods are effective and widely used, their primary limitation lies in their computational inefficiency. Iterative fitting requires multiple passes through the data, often with additional refinements to incorporate spatial priors and baseline corrections. As a result, traditional methods such as the FITT algorithm from the Metabolite Imaging and Data Analysis System (MIDAS) package [11] take an average of 45 min per scan to complete. This makes these methods impractical for clinical workflows where rapid data processing is critical.

NNFit addresses this limitation by replacing iterative fitting with a deep learning approach that generates results in a single forward pass [12]. This allows NNFit to process data in approximately 15 s per scan, offering a dramatic reduction in computational time while maintaining comparable accuracy to iterative methods. By overcoming this barrier, NNFit enables the practical integration of MR spectroscopy quantification into time-sensitive clinical workflows, such as radiation therapy planning.

In recent years, deep learning methods for spectral quantification have demonstrated their potential in addressing the limitations of conventional approaches [13,14,15,16,17]. These methods have primarily been applied to single-voxel spectroscopy (SVS) datasets, leveraging simulated spectra for training. While effective in controlled scenarios, their applicability to in vivo data remains constrained due to the absence of robust mechanisms for handling variability in spectral quality, phase, and frequency shifts. Moreover, existing methods are not optimized for high-resolution EPSI datasets, which generate 3D metabolite maps across large brain volumes and present additional computational challenges. This study builds on prior work by introducing NNFit, a method uniquely tailored to high-resolution EPSI data. NNFit addresses the limitations of prior methods by enabling self-supervised training directly on in vivo EPSI data, bypassing the need for simulation-based training and offering a robust solution for high-resolution metabolite quantification.

## 2. Materials and Methods

### 2.1. Clinical Datasets 

This study used two primary datasets of GBM patients undergoing sMRI as part of their treatment. The first dataset was collected from a clinical trial investigating sMRI-guided RT in GBM patients (ClinicalTrials.gov Identifier: NCT03137888). It consisted of 38 subjects who had one or more echo-planar spectroscopic imaging (EPSI) scans, resulting in a total of 68 scans. Patients were recruited from Emory University, the University of Miami, and Johns Hopkins University. For further details on the trial, refer to [6].

The second dataset came from a pilot study at the University of Miami, comprising 9 glioblastoma patients and 17 scans. This study was IRB-approved, and data transfer followed institutional data-sharing agreements. 

### 2.2. Image Acquisition 

For both datasets, imaging was conducted on Siemens 3T scanners using either a 32-channel or 20-channel head and neck coil. An EPSI pulse sequence with GRAPPA parallelization was used [18], with an echo time (TE) of 50 ms, repetition time (TR) of 1551 ms, and a flip angle (FA) of 71°. The field of view (FOV) was 280 mm × 280 mm × 180 mm with a matrix size of 64 × 64 × 32, yielding an interpolated voxel volume of 108 μL. Each scan took 14.5 min. During the same session, a non-contrast T1-weighted MRI with 1 mm isotropic voxels was acquired. Data from both the EPSI and T1-weighted sequences were processed using the MIDAS software suite (University of Miami, Miami, FL, USA). 

### 2.3. Dataset Split 

The first dataset was randomly split at the subject level into a training set (30 subjects, 56 scans) and a testing set (8 subjects, 12 scans). The second dataset was entirely used for testing, yielding a final test set of 17 subjects (29 scans). The training set contained approximately 450,000 spectra, while the test set included around 230,000 spectra. The final test set comprised subjects from Emory (4), Miami (11), and Johns Hopkins (2). Figure 1 illustrates the data flow and dataset split. 

### 2.4. Data Pre-Processing and Augmentation 

MIDAS was used to segment the normal-appearing white matter (NAWM) on the contralateral side to the tumor in the T1-weighted images [19]. Tissue segmentation (into CSF, gray matter, and white matter) and nonlinear registration to an anatomical atlas were performed using FSL v6.0.3 (https://fsl.fmrib.ox.ac.uk/, accessed on 1 August 2022). From the atlas, masks for the cerebellum and brainstem were generated, with NAWM, CSF, cerebellum, and brainstem masks used in later stages. 

The spectral quantification of sMRI data was performed using the FITT algorithm in MIDAS, which uses linear combination modeling and incorporates information from quantum–mechanical simulations generated by Vespa [20] for metabolites Cho, Cr, and NAA [11]. The MIDAS software (QMap) was then applied for quality control, filtering spectra based on signal-to-noise ratio and linewidth to remove low-quality spectra. QMap removed spectra with poor quality across the dataset. 

### 2.5. NNFit Network Architecture 

The NNFit model was designed to accelerate the spectral quantification of sMRI data [12]. The current architecture is composed of two key components, encoding networks and decoding networks, with one set focused on metabolites and the other on spectral baseline modeling (Figure 2). The encoders used a 1D-ResNet18 architecture with 16 filters in each convolutional layer in the starting block, with the number of filters doubling in subsequent blocks, following the standard ResNet convention. The metabolite encoding network is responsible for extracting features related to metabolite concentrations from individual spectra. The baseline encoding network focuses on modeling the non-metabolite components of the spectra. Both encoding networks output a vector in a lower-dimensional latent space, capturing the essential information required for spectral modeling.

The metabolite decoder network estimates the concentrations of key metabolites, including Cho, Cr, and NAA. This network uses a spectral model based on a basis set generated by Vespa, as is used in the FITT algorithm. A fully connected layer maps the latent vector from the metabolite encoder to the parameters of the spectral model.

The baseline decoder network models the underlying baseline spectra using a wavelet model. This model accounts for non-metabolite components in the spectra, such as macromolecule contributions and baseline distortions, improving the accuracy of metabolite quantification. By separating the baseline from the metabolite signals, this decoder helps NNFit focus on the relevant spectral information. A fully connected layer maps the latent vector from the baseline encoder onto the wavelet coefficients of the wavelet model.

### 2.6. Training 

NNFit was trained on ~450,000 spectra selected from the 56 scans in the training set, with data augmentation through random phase shifts (from −180° to 180°) and frequency shifts (from +20 Hz to −20 Hz). Since no ground-truth metabolite concentrations were available for in vivo spectra, the model was trained in a self-supervised way, minimizing mean square error (MSE) between estimated and original spectra. The model was implemented using the TensorFlow deep learning framework. Training was carried out on an NVIDIA 3080 Ti GPU (NVIDIA, Santa Clara, CA, USA) with 12 GB of memory using a learning rate of 1 × 10^−4^, a batch size of 512, and 200 epochs. 

### 2.7. Inference 

After training, NNFit generated spectral model parameter maps from sMRI scans in the test dataset. By processing spectra in batch forward passes, NNFit reduced computation time to 15 s per sMRI scan. This efficiency supports clinical use, providing rapidly computed, normalized metabolite ratio maps essential for treatment planning. 

The parameters determined by the NNFit analysis included metabolite estimates (Cho, Cr, NAA), frequency shifts, zero-order phase, and linewidth assuming a Gaussian decay. A map of the normalized ratio of Cho to NAA was then calculated to provide the maps from which the radiation treatment volumes were calculated. To avoid the ratio becoming arbitrarily large as the denominator approached zero, the denominator metabolite was thresholded at the lowest 5% of its distribution. This ensured that the ratio maps reflected biologically relevant variations rather than artifacts from near-zero metabolite concentrations.

Radiation treatment volumes were calculated following the protocol established in the clinical trial (NCT03137888) [6] from which Dataset #1 was obtained. For each scan, the average Cho/NAA ratio was computed in the contralateral normal-appearing white matter (NAWM) region. A mask was then created for regions where the Cho/NAA ratio exceeded twice that in NAWM (Cho/NAA ≥ 2x), and the primary connected component of this mask was extracted. Although the clinical trial involved the manual editing of these components by clinicians, in this study, an automated approach was used to provide an objective comparison between NNFit and MIDAS FITT. For radiation treatment volumes, a Cho/NAA ≥ 2x was used to define metabolically abnormal regions, with filtering steps to exclude voxels with poor quality, CSF, cerebellum, and brainstem regions. This automated approach, compared to manual adjustments in the clinical trial, provides a consistent basis for comparing NNFit with FITT. 

### 2.8. Statistical Analysis 

To evaluate NNFit against the MIDAS FITT algorithm, we used the structural similarity index (SSIM), Pearson correlation, and Dice coefficient [21,22,23]. SSIM quantified similarity in metabolite maps, Pearson correlation evaluated linear relationships, and the Dice coefficient measured overlap in radiation treatment volumes. Together, these metrics offered a comprehensive comparison between NNFit and current standard FITT, which rationalizes the potential replacement of FITT with NNFit in sMRI processing. 

While statistical measures like confidence intervals or *p*-values were not included, the primary goal of this study was to provide descriptive metrics for agreement rather than hypothesis testing. Confidence intervals or *p*-values are typically used when a clear reference standard or ground truth exists, which is not the case in this study, since both NNFit and FITT estimate metabolite concentrations and treatment volumes without an absolute reference.

In addition to using the Structural Similarity Index Measure (SSIM) to evaluate the metabolite maps, we included Bland–Altman plots to assess agreement between NNFit and FITT for each metabolite on the entire test dataset. The Bland–Altman plots provide a visual representation of the differences between methods across the range of metabolite values, highlighting any systematic bias or variability. By incorporating both SSIM and Bland–Altman plots, we provide a more comprehensive evaluation of the agreement between the two methods.

### 2.9. Integration with Clinical Studies 

NNFit is integrated into the clinical workflow through the Brain Imaging Collaboration Suite (BrICS), a cloud-based platform for incorporating sMRI with standard imaging. BrICS allows the visualization of metabolite and ratio maps, overlaid on anatomical MRIs, and supports multi-institutional collaboration for radiation therapy (RT) planning [19]. BrICS enables clinicians to define treatment volumes based on metabolite thresholds (e.g., Cho/NAA ratio), with options for manual RT volume adjustments. The rapid processing of NNFit, combined with BrICS’s collaborative features, provides an efficient and accurate platform for analyzing sMRI data in clinical trials. 

## 3. Results

### 3.1. Spectral Fits 

The NNFit model completed the spectral analysis of thousands of spectra in a volumetric sMRI dataset in approximately 15 s per scan—a significant improvement over the 45 min required by the FITT algorithm on a high-performance, multicore computer. Figure 3 shows spectral fits for four examples, with NNFit results (in blue) and FITT results (in orange) overlaid. Both models closely align in estimating key metabolite peaks for Cho (3.20 ppm), Cr (3.03 ppm), and NAA (2.01 ppm). Minor differences are observed between the Cr and NAA peak heights, which do not affect the overall metabolite estimates.

### 3.2. Metabolite Maps 

In Figure 4A, Bland–Altman plots have been generated for key metabolite quantifications, illustrating the differences between NNFit and FITT across the metabolite value range. The plots show that the differences were distributed within acceptable limits of agreement, with no significant systematic bias observed. These findings further support the robustness and reliability of NNFit in comparison to FITT.

In general, NNFit and FITT demonstrate near-perfect agreement. Figure 4B presents box-and-whisker plots of the structural similarity index (SSIM) between FITT and NNFit metabolite maps. The median SSIM values were as follows: 0.94 (IQR 0.93–0.95) for Cho, 0.94 (IQR 0.92–0.96) for Cr, 0.94 (IQR 0.92–0.95) for NAA, and 0.93 (IQR 0.90–0.97) for the Cho/NAA ratio, reflecting high similarity between the maps generated by both models.

### 3.3. Radiation Treatment Volumes 

To evaluate the clinical applicability of NNFit, we compared radiation treatment volumes based on Cho/NAA ≥ 2x between FITT and NNFit. Figure 5 displays the resultant high-dose radiation targets for six scans. Regions in red indicate where both models meet the 2x threshold; blue regions indicate areas where only FITT meets the threshold, and yellow regions indicate areas where only NNFit meets it. Overall, FITT and NNFit generate nearly identical contours for the 2x threshold, suggesting that treatment plans based on either model would be nearly identical. This close similarity is particularly relevant, as radiation treatment planning usually incorporates an additional margin to account for infiltrating tumor regions.

In Figure 6A, Cho/NAA ≥ 2x treatment volumes from FITT and NNFit are plotted for each scan in the test dataset. A correlation coefficient of 0.986 indicates a strong concordance between the models’ treatment volumes. Figure 6B shows the Dice similarity scores between FITT and NNFit treatment volumes, with a median Dice score of 0.87 (IQR 0.81–0.93), indicating substantial overlap and consistency in defining treatment regions. 

### 3.4. Clinical Translation 

For clinical validation, Figure 7 provides an example patient from the test dataset, visualized in the BrICS web platform. The figure presents the patient’s clinical images (T1w/CE, T2/FLAIR) alongside Cho/NAA maps from both FITT and NNFit. The NNFit map did not produce the visual artifact shown in the FITT map, while accurately capturing spectra within tumor regions (indicated by x’s), demonstrating NNFit’s ability to maintain clinical accuracy while improving image quality.

## 4. Discussion

This study evaluated the performance of NNFit, a deep learning-based spectral fitting model, against the established MIDAS FITT algorithm in metabolite quantification and treatment planning for patients undergoing radiation therapy. The findings demonstrate that NNFit offers a substantial improvement in computational efficiency while delivering comparable accuracy in key metabolite quantification and spatial mapping. These results suggest that NNFit has the potential to enhance the speed and feasibility of whole-brain 3D sMRI in clinical workflows, especially in time-sensitive scenarios like radiation treatment planning for GBM. 

NNFit introduces two key innovations that differentiate it from existing deep learning methods for metabolite quantification. First, NNFit leverages LCM principles to enable self-supervised training directly on in vivo EPSI datasets, overcoming the generalizability issues associated with training on simulated data. Second, NNFit applies robust phase and frequency correction (±180° phase, ±20 Hz frequency), ensuring accurate metabolite quantification.

The clinical relevance of NNFit lies in its ability to overcome a key barrier to the adoption of spectroscopic imaging in routine clinical workflows: processing time. Conventional methods, such as FITT, require approximately 45 min per scan, making their integration into time-sensitive workflows impractical. NNFit reduces this processing time to just 15 s while maintaining comparable accuracy in both metabolite quantification and treatment volume delineation. This significant reduction in time can enable the near-real-time availability of metabolic maps, allowing clinicians to incorporate the results into diagnostic and therapeutic decision-making during the same clinical session. By enabling the automated and efficient processing of EPSI data, NNFit addresses a critical need for advanced imaging techniques in oncology and other fields where metabolic abnormalities play a diagnostic and therapeutic role.

The close agreement between NNFit and FITT in estimating metabolite levels, especially for Cho, Cr, and NAA, underscores the reliability of NNFit for primary metabolite quantification. This level of accuracy is crucial because precise metabolic mapping supports more accurate tumor delineation and assessments of tumor heterogeneity, which are essential for guiding treatment strategies. The NNFit-generated metabolite maps also exhibited fewer visual artifacts, as seen in the Cho/NAA maps where FITT occasionally displayed hot spots. These hot spots/noisy spectra are often found in the periphery of the brain due to multiple causes, including the improper suppression of water/lipid signal and magnetic field inhomogeneities. These findings suggest that NNFit may reduce the number of incorrect fit results in areas of decreased spectral quality, thus improving map clarity, and potentially enhancing the diagnostic utility of sMRI by minimizing false-positive signals in areas not affected by disease. Such improvements can translate to greater confidence in metabolic quantification and, in turn, more reliable treatment targeting. The very high SSIM values for Cho, Cr, and NAA further support the clinical interchangeability of NNFit and FITT, assuring that NNFit can reliably replace FITT for these applications.

One of the most impactful findings of this study was the high correlation between FITT and NNFit in defining high-dose radiation targets based on Cho/NAA ≥ 2x thresholds. The observed Dice similarity score (median of 0.87) indicates substantial overlap in treatment volumes, considering 3–5 mm margins routinely added to the target volume, suggesting that either method could be used to define clinically relevant treatment regions without significantly altering patient treatment plans. This is especially promising in RT planning, where precision is critical, and NNFit’s accelerated processing could reduce planning time and expedite the treatment process, which is beneficial for both patients and clinicians. The high correlation (r = 0.986) between the treatment volumes generated by FITT and NNFit also demonstrates that NNFit’s contouring is practically indistinguishable from FITT’s, even under stringent thresholds like Cho/NAA ≥ 2x. These findings support NNFit’s suitability for integration into radiotherapy workflows and highlight its potential for accelerating processes without compromising treatment efficacy. In settings where rapid treatment adjustments are required, such as adaptive radiotherapy, NNFit could be particularly valuable by allowing clinicians to quickly re-evaluate and optimize treatment areas based on up-to-date metabolic information. 

While this study demonstrates that NNFit achieves comparable results to FITT in terms of metabolite quantification and radiation treatment volume delineation, the assumption that the clinical outcomes of both methods are equivalent requires further validation. Specifically, we plan to compare Cho/NAA treatment volumes in a prospective ECOG-ACRIN national trial to confirm the clinical equivalence of both fitting methods.

This study’s retrospective analysis provides strong preliminary evidence by showing high agreement between NNFit and FITT in metabolite maps and radiation treatment volumes. However, prospective studies involving clinical decision-making and patient outcomes are an essential next step to confirm the clinical equivalence of the two methods, and to further establish NNFit’s reliability and utility in practice.

The ability of NNFit to provide rapid, accurate, and artifact-reduced spectral fitting represents a major advance in metabolic imaging and radiotherapy planning. By decreasing the processing time to seconds, NNFit facilitates on-scanner reconstruction for metabolic imaging, paving the way for clinical integration and more dynamic, adaptive treatment strategies. This capability is especially relevant as personalized precision medicine becomes increasingly prominent in oncology. With continued refinement and broader validation, NNFit could significantly streamline clinical workflows, reduce patient wait times, and allow for more agile treatment planning. 

In conclusion, NNFit’s rapid processing that is comparable to the standard analysis method, and its artifact reduction, make it a promising tool for the clinical translation of 3D whole-brain spectroscopic MRI. Its potential impact on radiation treatment planning, coupled with its compatibility with platforms like BrICS, highlights its readiness for integration into routine clinical practice. Future research should focus on expanding its application across broader datasets, such as short TE, to model additional peaks for use as biomarkers for various neurological diseases, in order to fully realize its potential as a transformative tool in the clinical setting beyond GBM. 

## Figures and Tables

**Figure 1 cancers-17-00423-f001:**
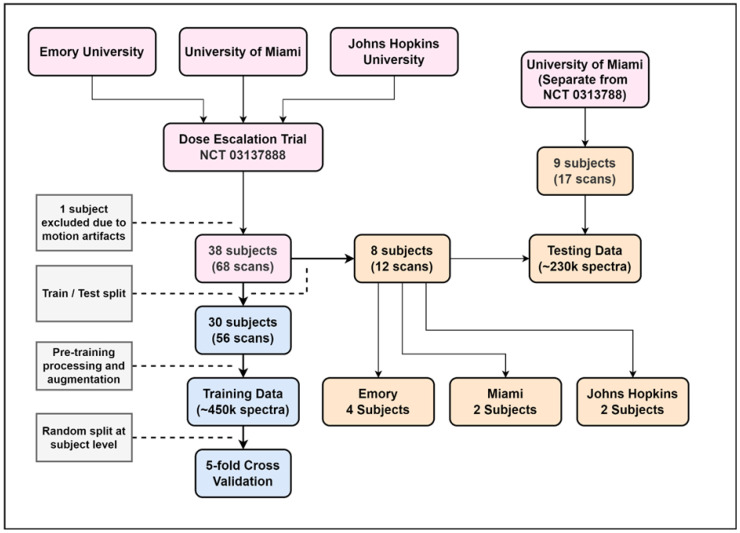
Data flow for patients and spectra used to train and test the NNFit model. Of note, sMRI acquisitions used TE = 50 ms.

**Figure 2 cancers-17-00423-f002:**
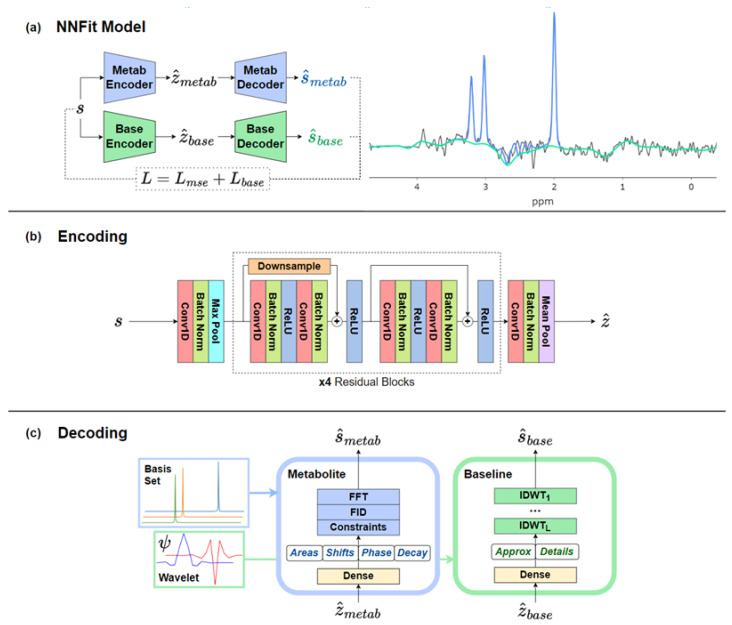
The NNFit model architecture (**a**) uses two sets of encoder and decoder models to learn relevant features and generate baseline (green) and metabolite (blue) fits. (**b**) A 1D-Resnet18 architecture was used for both the metabolite and baseline encoders in order to extract features from the data. (**c**) The metabolite decoder uses a linear combination model to estimate the metabolite signal, while the baseline decoder uses a wavelet model to estimate the baseline. The processing in the decoders includes the Fast Fourier Transform (FFT), Free Induction Decay (FID), and Inverse Discrete Wavelet Transform (IDWT). The FID represents the raw time-domain signal collected during MR spectroscopy, the FFT converts this time-domain signal into the frequency domain for spectral analysis, and the IDWT reconstructs the baseline signal from wavelet coefficients. Dense layers are used to map the latent vectors onto the parameters of the decoding models.

**Figure 3 cancers-17-00423-f003:**
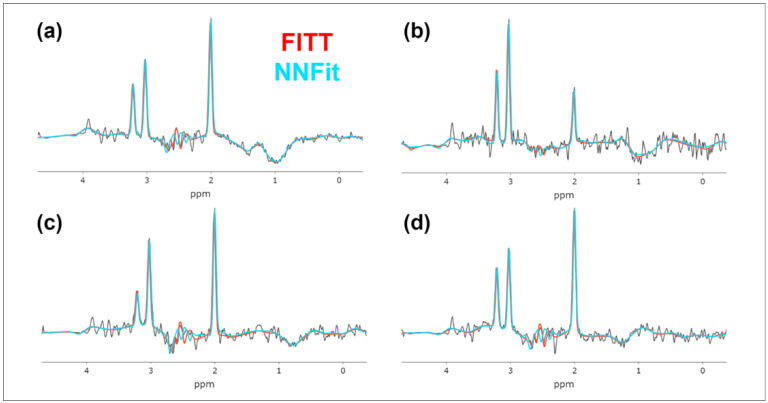
A comparison of MIDAS FITT and NNFit. (**a**–**d**) The spectral fittings for Cho (3.20 ppm), Cr (3.03 ppm), and NAA (2.01 ppm) are in perfect agreement.

**Figure 4 cancers-17-00423-f004:**
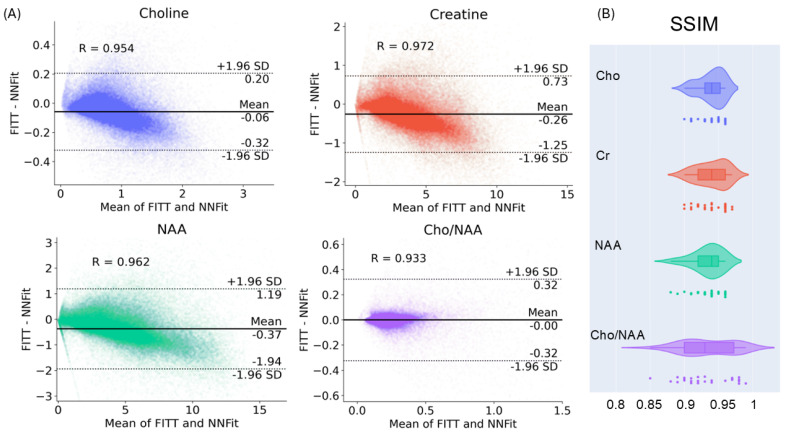
(**A**) The Bland–Altman plots comparing FITT and NNFit show minimal bias and acceptable agreement. (**B**) Box-and-whisker plots show a majority of SSIM values above 0.9 when comparing metabolite maps generated by FITT and NNFit.

**Figure 5 cancers-17-00423-f005:**
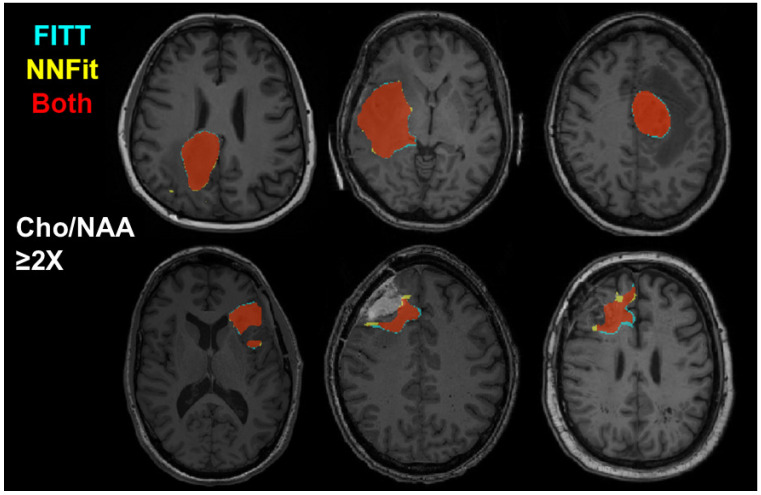
Using the Cho/NAA ≥ 2x, threshold, we generate representative examples of radiation treatment planning contours for six patients. FITT and NNFit have near-perfect agreement when translating metabolite maps to treatment plans.

**Figure 6 cancers-17-00423-f006:**
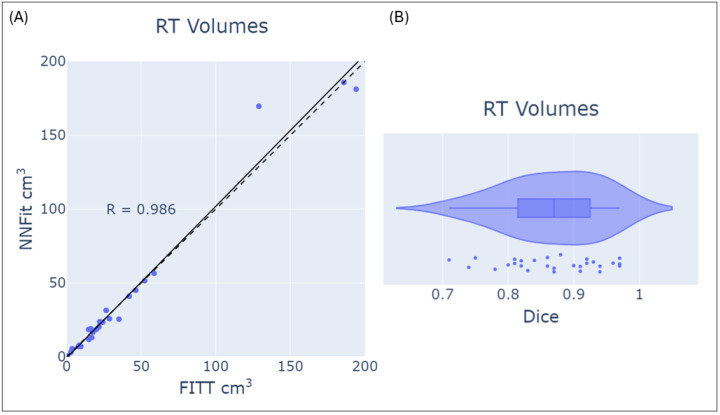
A comparison of RT volumes generated by FITT and NNFit using the Cho/NAA ≥ 2x threshold. (**A**) Plots the FITT vs. NNFit RT volumes for each scan in the test dataset with a correlation coefficient of 0.986 from the line of best fit while (**B**) shows a box-and-whisker of the Dice correlation coefficient between the volumes in the test dataset.

**Figure 7 cancers-17-00423-f007:**
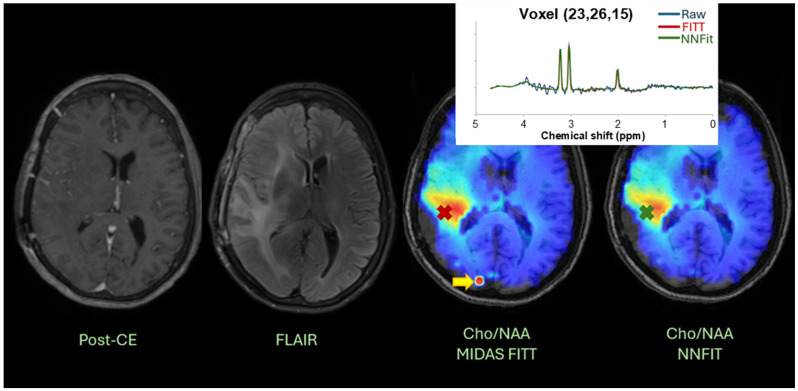
An example patient in the BrICS web platform where NNFit has been integrated successfully. In this case, we see an artifact in the MIDAS Cho/NAA map that is not visible in the NNFit map. A spectra graph is overlaid showing a comparison of FITT and NNFit for a voxel in the tumor region showing near agreement.

## Data Availability

To access these datasets, external researchers must establish a Data Transfer Agreement by contacting the corresponding author. To access the codebase, external researchers must establish a Material Transfer Agreement by contacting the corresponding author.

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
