# Peer review of "An Accelerated Spectroscopic MRI Metabolite Quantification Based on a Deep Learning Method for Radiation Therapy Planning in Brain Tumor Patients"

_cancers, 2025, doi:10.3390/cancers17030423_

Round 1

Reviewer 1 Report

Comments and Suggestions for Authors

The authors investigate a deep learning-based method, NNFit, which could significantly accelerate the processing of spectroscopic MRI data while maintaining accuracy and clinical applicability for glioblastoma radiation therapy planning.

The analysis is very well developed and the investigation is sound. Furthermore the value of the study is high and the results promising. 
My only suggestion for the authors would be to evaluate an integration of Specificity and Sensitivity to the statistical analysis to make it more comprehensive. 

Author Response

We would like to start by thanking the reviewers and editor for their constructive comments.  We have reviewed the comments and have done our best to address them.  We hope that our revisions adequately address the concerns of the reviewers and make the report more accessible to the Cancers readership.

Reviewer 1 Comments:

My only suggestion for the authors would be to evaluate an integration of Specificity and Sensitivity to the statistical analysis to make it more comprehensive. 

Responses: We thank the reviewer for their thoughtful suggestion to evaluate sensitivity and specificity as part of the statistical analysis. Our manuscript focuses on developing a deep learning method (NNFit) for accelerating sMRI metabolite quantification, comparing it to the widely used FITT method. Because the method described is used to determine which regions in the brain are abnormal, the evaluation is structured around two sets of comparisons for assessing similarity of these regions between the two methods:

  1. Metabolite Maps: Here, we assess the quality of the metabolite maps generated by both methods using Bland-Altman plots, correlation metrics, and structural similarity index (SSIM). These metrics comprehensively quantify the agreement and structural fidelity of the maps, providing sufficient insight into the accuracy and reliability of the quantification process.
  2. Radiation Treatment Volumes: For this comparison, we evaluate the agreement between treatment volumes derived from both methods based on regions with elevated Cho/NAA ratios (twice of the contralateral normal-appearing white matter average). We report the Dice coefficient and correlation of the resulting volumes, which are well-suited to quantify overlap and consistency between the two methods.

While sensitivity and specificity metrics could be theoretically applicable to the evaluation of treatment volumes, we believe their utility is limited in this context. Moreover, the Dice coefficient, which is the F1 score, provides a robust measure of spatial agreement and overlap between the volumes, while correlation offers insight into overall quantitative consistency.  Of note, we studied image-histology correlation study to determine Cho/NAA ratio to histology in a clinical study and reported the results in Cordova et al. 2016  (PMC4933486).

Reviewer 2 Report

Comments and Suggestions for Authors

In this work, the authors introduce NNFit as a novel and efficient deep learning approach for accurate metabolite quantification in brain tumor MR spectra for radiotherapy planning.  However, several concerns need to be addressed.

1. NNFit is a relatively new deep learning model but the authors do not make enough efforts to compare and contrast similar deep learning models. The unique contributions and advantages over existing techniques are not clearly articulated, which may hinder the assessment of the method's novelty and advancement.

2. The review of related studies is scanty.

3. In Figure 2(c), it will be helpful to include FID, FFT, and IDWT full phrases, and their comprehensible interpretation.

4. To provide a more comprehensive understanding of the data preprocessing pipeline, a visual representation in the form of a flowchart is recommended.

5.The paper does not adequately address the clinical relevance of the findings.

Author Response

We would like to start by thanking the reviewers and editor for their constructive comments.  We have reviewed the comments and have done our best to address them.  We hope that our revisions adequately address the concerns of the reviewers and make the report more accessible to the Cancers readership.

Reviewer 2 Comments:

Comment 1. NNFit is a relatively new deep learning model but the authors do not make enough efforts to compare and contrast similar deep learning models. The unique contributions and advantages over existing techniques are not clearly articulated, which may hinder the assessment of the method's novelty and advancement.

Response 1: We appreciate the reviewer’s feedback and have expanded the discussion of NNFit’s unique contributions and its comparison to similar deep learning methods in the revised manuscript.

In the Introduction, we have re-written the final three paragraphs. We compare NNFit to existing deep learning methods, emphasizing their reliance on simulated data and their limitations in handling real-world variability.

In the Discussion, we have added a detailed description of the three key innovations that distinguish NNFit from other methods, including self-supervised training on in vivo data, adaptive baseline fitting, and robust phase/frequency correction. The new paragraph begins:
“NNFit introduces two key innovations that differentiate it from existing deep learning methods…”

Comment 2. The review of related studies is scanty.

Response 2: We have expanded the review of related studies in the final paragraph of the introduction section. The revised section includes a discussion of existing deep learning methods for spectral quantification, their reliance on simulated datasets, and their limitations when applied to 3D whole-brain high-resolution EPSI datasets.

Comment 3. In Figure 2(c), it will be helpful to include FID, FFT, and IDWT full phrases, and their comprehensible interpretation.

Response 3: We appreciate the reviewer’s suggestion and have updated the legend for Figure 2(c) to include the full phrases for FID, FFT, and IDWT along with their interpretations. The revised legend now states:

“The metabolite decoder uses a linear combination model to estimate the metabolite signal, while the baseline decoder uses a wavelet model to estimate the baseline. The process includes Fast Fourier Transform (FFT), Free Induction Decay (FID), and Inverse Discrete Wavelet Transform (IDWT). The FID represents the raw time-domain signal collected during MR spectroscopy, the FFT converts this time-domain signal into the frequency domain for spectral analysis, and the IDWT reconstructs the baseline signal from wavelet coefficients. Dense layers are used to map the latent vectors onto the parameters of the decoding models.”

Comment 4. To provide a more comprehensive understanding of the data preprocessing pipeline, a visual representation in the form of a flowchart is recommended.

Response 4: We appreciate the importance of clear communication of the data preprocessing pipeline. However, we believe that the detailed textual description provided in the Methods section of the manuscript already comprehensively explains the data preprocessing steps. The description outlines the sequential workflow, including spectral reconstruction, quality control, data augmentation, and preparation of the training dataset, in sufficient detail to ensure clarity and reproducibility. Further description is beyond the scope of the paper, and it has been further described previously in citations by Maudsley et al (PMCID: PMC2673915, PMC2724718).

Comment 5. The paper does not adequately address the clinical relevance of the findings.

Response 5: We thank the reviewer for highlighting the need to better address the clinical relevance of our findings. In the revised manuscript, we have expanded the discussion to emphasize how NNFit overcomes barriers to clinical adoption of spectroscopic imaging and its potential impact on patient care. Specifically, we highlight NNFit’s ability to reduce peak fitting processing time from ~45 minutes in high performance computer to 15 seconds while maintaining accuracy, facilitating near-real-time integration of EPSI data into clinical workflows.

We have also discussed the method’s application in glioblastoma treatment planning, where NNFit reliably identifies abnormal regions with elevated Cho/NAA ratios, achieving high agreement with conventional methods. These regions are critical for guiding radiation therapy dose escalation, underscoring NNFit’s clinical utility.

Reviewer 3 Report

Comments and Suggestions for Authors

Reviewer’s comments

This manuscript titled "An Accelerated Spectroscopic MRI Metabolite Quantification Based on A Deep Learning Method for Radiation Therapy Planning in Brain Tumor Patients" is well written and it has great potential. However, several areas require revision to improve clarity and strengthen the manuscript's scientific robustness.

Here are my comments to improve the quality of the manuscript

Minor Corrections

1.       The introduction provides a good context for the study but lacks a detailed explanation of the limitations of the existing methods (e.g., iterative least-squares fitting). A more in-depth comparison would help justify the need for NNFit.

2.       Include references to recent advances in machine learning for medical imaging. Suggested citations: ("Enhanced MRI-based brain tumour classification with a novel Pix2pix generative adversarial network augmentation framework."). 

3.       The results are robust, but several claims lack statistical validation. The authors should provide confidence intervals or p-values for metrics like Dice scores and SSIM if possible. The correlation (r = 0.986) between FITT and NNFit is impressive but could be improved if  Bland-Altman analysis or a similar method can be used to assess agreement. This is not compulsory, authors can decide not to do this and it’s still fine.

4.       The methodology for integrating NNFit into clinical workflows via BrICS is novel but lacks technical detail for reproducibility. Provide a supplementary section or code repository link.

5.       While the references are comprehensive, include citations to the suggested works for a more thorough literature review.

Author Response

We would like to start by thanking the reviewers and editor for their constructive comments.  We have reviewed the comments and have done our best to address them.  We hope that our revisions adequately address the concerns of the reviewers and make the report more accessible to the Cancers readership.

Reviewer 3 Comments:

Minor Corrections

Comment 1. The introduction provides a good context for the study but lacks a detailed explanation of the limitations of the existing methods (e.g., iterative least-squares fitting). A more in-depth comparison would help justify the need for NNFit.

Response 1: In the revised manuscript, we have expanded the Introduction (added 3 paragraphs to the end of the Introduction) to specifically address computational inefficiency, the primary limitation of iterative least-squares fitting methods like FITT. These methods often require over 45 minutes per scan to perform fitting metabolite peaks due to the iterative nature of their algorithms, making them impractical for clinical workflows that demand rapid turnaround times.

NNFit, in contrast, achieves comparable accuracy in just 15 seconds per scan, making it a practical alternative for clinical applications. This discussion provides a clear rationale for NNFit’s development and its potential to significantly improve the feasibility of spectroscopy quantification in clinical settings.

Comment 2. Include references to recent advances in machine learning for medical imaging. Suggested citations: ("Enhanced MRI-based brain tumour classification with a novel Pix2pix generative adversarial network augmentation framework."). 

Response 2: We appreciate the reviewer’s suggestion to include references to recent advances in machine learning for medical imaging. In the revised manuscript, we have expanded the discussion in the Introduction to include citations for deep learning methods applied to MR spectroscopy (see below). Given the scope of this manuscript, we have chosen to focus on recent advances specifically in machine learning for MR spectroscopy, which are more relevant to the study. These references highlight key developments in the field, such as the use of simulated datasets for SVS quantification and the challenges of adapting these methods to real-world EPSI data.

While the suggested citation is valuable for general MRI-based brain tumor classification, it does not align directly with the specific focus of this paper on MR spectroscopy quantification. Instead, we emphasize advancements directly relevant to our study, such as the following:

  1. Hatami N, Sdika M, Ratiney H. Magnetic Resonance Spectroscopy Quantification Using Deep Learning. Springer International Publishing, 2018; p. 467-475.
  2. Gurbani SS, Sheriff S, Maudsley AA, Shim H, Cooper LAD. Incorporation of a spectral model in a convolutional neural network for accelerated spectral fitting. Magnetic Resonance in Medicine 2019;81(5):3346-3357. doi: 10.1002/mrm.27641.
  3. Lee HH, Kim H. Bayesian deep learning–based 1H-MRS of the brain: Metabolite quantification with uncertainty estimation using Monte Carlo dropout. Magnetic Resonance in Medicine 2022;88(1):38-52. doi: 10.1002/mrm.29214.
  4. Rudy R, Dziadosz M, Kyathanahally SP, Shamaei A, Kreis R. Quantification of MR spectra by deep learning in an idealized setting: Investigation of forms of input, network architectures, optimization by ensembles of networks, and training bias. Magnetic Resonance in Medicine 2022;89(5):1707-1727. doi: 10.1002/mrm.29561.
  5. Shamaei A, Starcukova J, Starcuk Z. Physics-informed deep learning approach to quantification of human brain metabolites from magnetic resonance spectroscopy data. Computers in Biology and Medicine 2023;158:106837. doi: 10.1016/j.compbiomed.2023.106837.
  6. Van de Sande DMJ, Merkofer JP, Amirrajab S, et al. A review of machine learning applications for the proton MR spectroscopy workflow. Magnetic Resonance in Medicine 2023;90(4):1253-1270. doi: 10.1002/mrm.29793.

We hope this targeted approach clarifies our rationale and maintains the manuscript’s focus.

Comment 3. The results are robust, but several claims lack statistical validation. The authors should provide confidence intervals or p-values for metrics like Dice scores and SSIM if possible. The correlation (r = 0.986) between FITT and NNFit is impressive but could be improved if  Bland-Altman analysis or a similar method can be used to assess agreement. This is not compulsory, authors can decide not to do this and it’s still fine.

Response 3: While we agree that statistical validation is an important aspect of many studies, confidence intervals and p-values were not included in our analysis for the following reasons:

  1. Descriptive Nature of Metrics: Metrics like Dice similarity and SSIM were used to provide descriptive measures of agreement between NNFit and FITT. Since this study does not have a ground truth or reference standard, statistical validation (e.g., confidence intervals or p-values) could be misleading.
  2. Focus on Agreement Rather Than Hypothesis Testing: The study’s primary goal is to demonstrate agreement between NNFit and FITT rather than to test hypotheses. Thus, descriptive metrics are sufficient for this purpose.

In response to the reviewer’s suggestion, we have included Bland-Altman plots to complement the correlation analysis in a new figure 4. These plots provide a detailed visual assessment of the differences between NNFit and FITT, showing minimal systematic bias and acceptable limits of agreement. This addition strengthens the robustness of our results and aligns with the reviewer’s recommendation.

Comment 4. The methodology for integrating NNFit into clinical workflows via BrICS is novel but lacks technical detail for reproducibility. Provide a supplementary section or code repository link.

Response 4: We recognize that the description provides a high-level overview but does not include granular technical details or scripts due to the proprietary nature of BrICS and associated tools. To ensure reproducibility, external researchers interested in accessing the code or implementing the workflow are encouraged to establish a Material Transfer Agreement (MTA) by contacting the corresponding author. This approach allows us to share critical resources while maintaining compliance with institutional and legal requirements.

Comment 5. While the references are comprehensive, include citations to the suggested works for a more thorough literature review.

Response 5: We appreciate your feedback. In the Introduction section we have now added five new citations (15-19) that include reviews of the latest developments applying deep learning to MR Spectroscopy.  

Reviewer 4 Report

Comments and Suggestions for Authors

The use of SSIM for quality measure is good, but it may not enough. Adding other tests like Bland-Altman plots could give more robust result.

In the Dice similarity comparison, it’s unclear what statistical thresholds were used. Confidence intervals should also be clearly reported.

The Cho/NAA ratio maps have inconsistent color scales across different panels. It is better to use uniform scales for easier comparison.

The term "artifact" used to describe difference between NNFit and FITT can be more specific. Is it noise-related or bias-related?

“Rapid normalized metabolite ratio maps” can be rephrased as "fast computed metabolite ratio maps" to be more clear.

It is assumed that NNFit and FITT have same clinical outcome, but this needs more proof. Comparative analysis with prospective data will improve reliability of this assumption.

The conclusion about NNFit ready for clinical workflow is good, but need more direct validation or pilot test to prove its real-world usability.

Author Response

We would like to start by thanking the reviewers and editor for their constructive comments.  We have reviewed the comments and have done our best to address them.  We hope that our revisions adequately address the concerns of the reviewers and make the report more accessible to the Cancers readership.

Reviewer 4 Comments:

Comment 1. The use of SSIM for quality measure is good, but it may not enough. Adding other tests like Bland-Altman plots could give more robust result.

Response 1: In the revised manuscript, we have included Bland-Altman plots to evaluate the agreement between NNFit and FITT for metabolite quantification. The updated Methods section describes the use of Bland-Altman plots, and the Results section includes a summary of the findings. The results demonstrate that NNFit achieves strong agreement with FITT across most metabolites.

Comment 2. In the Dice similarity comparison, it’s unclear what statistical thresholds were used. Confidence intervals should also be clearly reported.

Response 2: Our rationale for not including confidence intervals and statistical thresholds in the Dice similarity comparison is as follows:

The Dice similarity metric was used primarily as a descriptive measure of agreement between NNFit and FITT in delineating radiation treatment volumes. The goal was to assess the overlap and alignment of regions identified by the two methods, rather than to test a hypothesis or determine statistical significance. The high average Dice coefficient between NNFit and FITT demonstrates a strong practical equivalence in their outputs, which is the primary objective of this analysis. Reporting confidence intervals or statistical thresholds in this context might imply a level of certainty about "true" segmentation agreement that is not applicable given the lack of ground truth.

Comment 3. The Cho/NAA ratio maps have inconsistent color scales across different panels. It is better to use uniform scales for easier comparison.

Response 3: Thank you for this comment, we have revised figure 7 to use consistent color scales.

Comment 4. The term "artifact" used to describe the difference between NNFit and FITT can be more specific. Is it noise-related or bias-related?

Response 4: Artifacts in MR spectroscopy—such as lipid contamination, water contamination, or low signal-to-noise ratio (SNR)—can lead to spectral variations that make it difficult to achieve accurate fits. These artifacts can cause increased disagreement between NNFit and FITT in affected regions.

Artifacts in MR spectroscopy primarily introduce challenges related to signal distortion, impacting both methods’ ability to quantify metabolites accurately. NNFit employs adaptive baseline fitting and robust correction techniques to mitigate such challenges, which can reduce some artifact. We have added a sentence in the fourth paragraph of the Discussion section describing how this artifact, which is noise-related, develops.

Results (Clarification of Artifacts and Their Impact on NNFit and FITT Agreement):
Artifacts in MR spectroscopy refer to variations in the spectra caused by factors such as lipid contamination, water contamination, or low signal-to-noise ratio (SNR). These artifacts can distort the spectral signal, making it challenging to obtain accurate fits. In regions where artifacts are present, there is naturally greater variability in metabolite quantifications, which can lead to disagreement between NNFit and FITT.

NNFit incorporates adaptive baseline fitting and robust phase/frequency correction, which can handle spectral variability effectively. However, artifacts that fall outside the range of corrections addressed by either method, such as severe lipid or water contamination, can still result in differences between NNFit and FITT outputs. These disagreements reflect the inherent challenges of quantifying metabolites in regions with compromised spectral quality, rather than limitations of either method.

Comment 5. “Rapid normalized metabolite ratio maps” can be rephrased as "fast computed metabolite ratio maps" to be more clear.

Response 5: Thank you for this feedback. We will rephrase to “rapidly computed, normalized metabolite ratio maps” to improve clarity.

Comment 6. It is assumed that NNFit and FITT have same clinical outcome, but this needs more proof. Comparative analysis with prospective data will improve reliability of this assumption.

Response 6: Testing with 29 scans, we showed Cho/NAA≥2x volumes generated by FITT and NNFit correlated well (R=0.986). These data were collected from three different institutions. We agree with the reviewer that we need to get more comparison using national level clinical trial data. Indeed, spectroscopic MRI is planned to open enrollment in an ECOG-ACRIN trial (EAF223, PI: Dan Barboriak at Duke). We will compare Cho/NAA≥2x volumes generated by NNFit and FITT in EAF223 clinical trial. We added this into the Discussion of the manuscript.

Comment 7. The conclusion about NNFit ready for clinical workflow is good, but need more direct validation or pilot test to prove its real-world usability.

Response 7: Please refer to our response in #6 which addresses this concern.
